# HIV-1 suppression and rare dolutegravir resistance in antiretroviral-experienced people with HIV in Liberia

James Soka Moses[1,2] ✉, Alice K. Pau[3], Safia Kuriakose[4], Greg Grandits[5], Cavan Reilly[5], Brad T. Sherman[6], Weizhong Chang[6], Lisheng Dai[6], Muhammad A. Khan[6], Helene Highbarger[6], Moses Mannah[1], Johnathan McCullough[7], Carla Chorley[8], Isaac Morlu[1], Joseph Dorbor[1], Ophelia Talweh Bongolee [1], Rebecca Slewion[1], Esther Akpa[8], Barthalomew Wilson[1], April L. Poole[3], Stacy L. Kopka[8], Tracey Miller[8], Cecelia J. Nuta[9], Christina Lindan[2], David Glidden[2], Jeffrey N. Martin [2], Kumblytee L. Johnson[1], Robin L. Dewar[6], Ian Wachekwa[1,9] & Stephen A. Migueles[3,10]

## Abstract

**Background** Increasingly, persons with HIV in Liberia are receiving antiretroviral therapy containing the integrase strand-transfer inhibitor (InSTI) dolutegravir (DTG), but the prevalence of and factors associated with virologic failure and HIV drug resistance (HIVDR) remain unknown.

**Methods** Cross-sectional analysis of 2019–2022 enrolment data from 1276 persons with HIV in the HONOR cohort included sociodemographic information, plasma viral loads (pVL), CD4 counts, and HIVDR testing by next generation sequencing in participants with virologic failure (pVL≥1000 copies/mL).

**Results** Of the 1201 participants with pVL results, 72% are female and median age is 42 (interquartile range [IQR] 35–50) years. All are on ART (median 6.1 [2.1–11] years): 74% on DTG-based and 23% on non-nucleoside reverse transcriptase inhibitor (NNRTI)-based regimens. Ninety (7.5%) had virologic failure; 970 (81%) are suppressed (<40 copies/mL). Virologic failure is less prevalent with DTG- versus NNRTI-based regimens (5.3% vs. 14%, adjusted prevalence ratio [aPR]=0.3, 95% confidence interval [CI] 0.2–0.5) and is associated with age <50 years, CD4 count <200 cells/μL, and hemoglobin <11 g/dL. In 70 participants with virologic failure and successful sequencing, HIVDR prevalence is 81% for any ARV, 5.7% for InSTIs, 79% for NNRTIs, and 61% for nucleos(t)ide reverse transcriptase inhibitors (NRTIs). Intermediate-to-high resistance to ≥1 NRTI in current ART is less prevalent with DTG+2NRTIs than NNRTI+2NRTIs regimens (aPR = 0.5, 95%CI 0.3–0.8).

**Conclusions** Most participants in the cohort are virologically-suppressed. Among those with virologic failure, HIVDR prevalence is high to NRTIs and NNRTIs, but low to InSTIs. Ongoing evaluation is necessary to determine the durability of DTG-based ART.

## Plain language summary

Antiretroviral therapy (ART) is a combination of medications used to treat people infected with human immunodeficiency virus (HIV). ART prevents HIV from replicating and so reduces the risk of a person infecting others with HIV. ART is recommended for all people with HIV in Liberia, but it is unknown how well people respond to treatment or whether the specific type of HIV they have is killed by ART. We measured the amount of HIV in some people on ART and whether the HIV they were infected with was killed by ART. Most people responded well to the treatment but people on ART containing the drug efavirenz were more likely than those taking dolutegravir not to respond. This information could be used to improve the combinations of medications used in ART to treat patients.

Antiretroviral (ARV) drug resistance among persons with HIV-1 (PWH) is a threat to ending the HIV epidemic[1,2]. Since early 2000s, non-nucleoside reverse transcriptase inhibitor (NNRTI)-based regimens have been recommended as first-line treatment for PWH in Africa. However, NNRTIs such as efavirenz (EFV) and nevirapine (NVP) are not optimal because of their high rates of toxicity and low genetic barriers to resistance. As of 2020,

an estimated 24% of adults and 45% of infants with HIV in low- or middle-income countries (LMIC) on antiretroviral therapy (ART) had NNRTI resistance[2]. In 2018, the World Health Organization (WHO) recommended switching first-line ART from an NNRTI-based regimen to the integrase strand transfer inhibitor (InSTI) dolutegravir (DTG) plus two nucleos(t)ide reverse transcriptase inhibitors (NRTIs)[3]. The WHO also recommended

that HIV programs conduct routine surveillance of population-level HIV drug resistance (HIVDR) to inform public health decision-making[2]. However, data on HIVDR are lacking for many countries in sub-Saharan Africa, which account for two-thirds of HIV cases globally. As of 2021, only 9 of the 54 countries in Africa, and none from West Africa, reported HIVDR data in adults to the WHO[2].

In Liberia, in the era of transition to DTG-based ART, prevalence of virologic failure and HIVDR have not been studied. Prior studies from Monrovia, which included small sample sizes and were conducted when plasma viral load (pVL) monitoring was unavailable, reported that 5.9% of ART-naïve patients, and >60% of those with viraemia while on first-line NNRTI-based therapy, had HIVDR[4,5]. In 2018, following WHO's recommendation, a fixed-dose combination of DTG, tenofovir (TDF), and lamivudine (3TC) was introduced as first-line therapy in Liberia. In addition, patients who were already on ART were switched to this regimen, regardless of pVL. Switching ART regimens without information on pVL or genotypic resistance profile may result in suboptimal treatment response. To date, despite an increasing proportion of Liberian PWH on ART (53% as of 2022)[6,7] and receiving pVL monitoring, challenges persist. Some clinics still lack resources for routine pVL testing, and drug resistance testing is unavailable[7,8]. In PWH with virologic failure, little is known in Liberia about the proportion with HIVDR and, consequently, the appropriate intervention. Achieving the third 95 (95% of ART recipients achieve virologic suppression) of the 2030 UNAIDS 95-95-95 global targets for ending the HIV epidemic requires identification and appropriate management of virologic failure[9].

In 2019, the U.S. National Institutes of Health (NIH)-sponsored Partnership for Research on Vaccines and Infectious Diseases in Liberia (PREVAIL) research program initiated a longitudinal study of PWH called CoHOrt Clinical, Viral, and ImmuNOlogic Monitoring Study of People Living with Retroviral Infection in Liberia (HONOR study). Using data obtained from participants at enrolment into HONOR, we investigated the prevalence and patterns of, and factors associated with, virologic failure and HIVDR. Herein, we report that the majority of ART-experienced participants (81%) are virologically suppressed. Virologic failure is associated with younger age, lower CD4 count, longer ART duration, lower hemoglobin levels, and current treatment with EFV, NVP, or AZT. A lower proportion of those on a DTG-based regimen had virologic failure. In those with virologic failure and successful virus sequencing, the prevalence of resistance is high and mainly to reverse transcriptase inhibitors. In contrast, resistance to InSTIs and PIs is low.

## Methods
### Design and study population
We conducted a cross-sectional study to investigate virologic failure and HIV genotypic drug resistance (GDR) in participants at the baseline visit for the HONOR study (ClinicalTrials.gov ID: NCT03733093). Participants represent a mixed demographic of PWH who were recruited from five of the largest ambulatory HIV clinics in Monrovia, Liberia, which provided care for 74% of the 12,600 PWH receiving care in Monrovia as of August 2022 (Liberia's National AIDS Control Program). Interested clinic attendees of any age and sex visited the PREVAIL research site at the John F. Kennedy Memorial Hospital (JFK) and were enrolled on study if they agreed to be followed by a participant tracker, allowed storage of biological samples for research testing, provided written informed consent (children 7–17 years old provided assent with parental consent) and underwent confirmatory testing of HIV-1 status (GS HIV Combo Ag/Ab EIA [Bio-Rad Life Science, Hercules, CA, USA] and Geenius HIV-1/2 Supplemental Assay [Bio-Rad Life Science] or plasma HIV-1 RNA detection [Xpert® HIV-1 Viral Load, Cepheid, Sunnyvale, California, USA]). The study was approved by the National Research Ethics Board, Liberia and the NIH Institutional Review Board, USA.

### Measurements
We collected self-reported data on sociodemographic characteristics and HIV history. When available, participants' medical records and treatment cards listing ARV history were reviewed by the study team and used to complete case report forms. Blood was drawn to measure pVL with a quantification limit of 40 copies/mL, CD4 count (BD FACSPresto™ System [BD Life Sciences, San Jose, California, USA]) hemoglobin, and D-dimer. Participants with a single HIV-1 pVL result at the baseline visit of <40 copies/mL were defined as having virologic suppression and those with ≥1000 copies/mL as having virologic failure.

### HIV resistance testing
Plasma samples for participants with pVL≥1000 copies/mL were stored in liquid nitrogen and shipped at −20 °C to the Frederick National Laboratory for Cancer Research, Frederick, Maryland, USA, to test for HIV GDR to 25 ARVs (8 protease inhibitors [PIs], 7 NRTIs, 5 NNRTIs, and 5 InSTIs). Viral RNA extraction, amplification, and next generation sequencing of the HIV-1 Gag-Pol region (about 4·5 kb) with MiSeq was performed[10,11]. We generated ambiguous consensus sequences at ≥5% and ≥15% frequency thresholds. The consensus sequences at the ≥15% frequency threshold align more closely with Sanger sequencing limitations. Each consensus sequence was submitted to the Stanford HIVDB (v9·4) drug resistance interpretation program through the Sierra Web Service 2 (https://hivdb.stanford.edu/page/webservice/) to characterize drug resistance mutations. Fewer drug resistance mutations were detected in ambiguous sequences at the ≥15% than the ≥5% frequency threshold. In this report, the 5% threshold data is being presented since using 15% threshold data could miss detection of important drug resistance mutations at virologic failure. The level of resistance to each drug was categorized as high, intermediate, low, or susceptible. Potential low-level resistance was not included in statistical analyses because the clinical significance of these mutations is unknown.

### Statistics and reproducibility
We used a fixed sample size; enrollment data from all participants in the HONOR Study with pVL measurements were used in this analysis. We report cohort baseline characteristics, stratified by pVL (<40, 40–999, and pVL≥1000 copies/mL) using proportions for categorical variables and median (interquartile range [IQR]) for continuous variables. Median pVL for the <40 copies/mL group was imputed at half the limit of detection, i.e., 20 copies/mL. We calculated unadjusted and adjusted prevalence ratios (aPR) and 95% confidence intervals (CI) for the association of individual ARVs at enrolment (EFV, NVP, TDF, lopinavir/ritonavir [LPV/r], and DTG) with virologic failure using logistic regression, comparing participants with virologic failure with participants with pVL<40 copies/mL. Marginal relative risks were calculated by regression standardization based on the logistic regression models using the *margins* command in Stata version 16[12,13]. The minimum sufficient adjustment set (age, sex, education level, hemoglobin, D-dimer, years on ART, and CD4 count), was determined using a Directed Acyclic Graph (DAG). Due to small numbers of missing values, a complete case analysis was conducted. The proportion of individuals with resistance to individual ARVs, ARV class, and ART regimen was computed for those in whom GDR testing was successful. The analysis was conducted with Stata version 17·0 (Stata Corp, College Station, TX, USA).

## Results
From August 2019 through March 2022, we enrolled 1276 PWH; 100% were on ART. Baseline pVLs were measured in 1201, but not in 75 participants (Supplementary Fig. 1) due to unsuccessful phlebotomy (*n* = 1) or delays related to batch processing of frozen plasma specimens (*n* = 74). Baseline characteristics were similar in those with or without pVL results (Supplementary Table 1). Of the 1201 with pVL results, 72% were female and median age was 42 (IQR 35–50) years (Table 1). Median time since HIV diagnosis was 6.6 (IQR 2.6–11) years, and median duration on ART was 6.1 (IQR 2.1–11) years. The most common regimens were DTG+2NRTIs (74%) and EFV/NVP+2NRTIs (23%). The median $\log_{10}$ pVL was 1.3 (IQR 1.3–2.1) copies/mL and median CD4 count was 558 (IQR 362–779) cells/μL (Table 1).

**Table 1 | Characteristics of persons with HIV on ART at enrolment into HONOR, by HIV plasma viral load, Liberia**

| | HIV plasma viral load (copies/mL) | | | |
| | <40 | 40–999 | ≥1000 | Total[a] |
| Characteristic | N = 970 | N = 141 | N = 90 | N = 1201 |
| --- | --- | --- | --- | --- |
| Age, years, median (IQR) | 43 (35–51) | 40 (34–47) | 38 (30–44) | 42 (35–50) |
| Age groups | | | | |
| <15 | 26 (70%) | 6 (16%) | 5 (14%) | 37 (3.1%) |
| 15–24 | 35 (65%) | 9 (17%) | 10 (19%) | 54 (4.5%) |
| 25–49 | 630 (79%) | 98 (12%) | 65 (8.2%) | 793 (66%) |
| ≥50 | 279 (88%) | 28 (8.8%) | 10 (3.2%) | 317 (26%) |
| Sex assigned at birth | | | | |
| Male | 261 (78%) | 49 (15%) | 25 (7.5%) | 335 (28%) |
| Female | 709 (82%) | 92 (11%) | 65 (7.5%) | 866 (72%) |
| Level of education[b] | | | | |
| None | 195 (81%) | 26 (11%) | 19 (7.9%) | 240 (20%) |
| Primary-high school | 586 (81%) | 82 (11%) | 55 (7.6%) | 723 (61%) |
| Vocational-University | 177 (80%) | 30 (14%) | 13 (5.9%) | 220 (19%) |
| BMI, kg/m$^2$, median (IQR)[b] | 24 (21–28) | 23 (21–28) | 23 (20–26) | 23 (21–28) |
| Hemoglobin (g/dL)[b,c] | | | | |
| ≥11 | 816 (84%) | 100 (10%) | 52 (5.4%) | 968 (81%) |
| <11 | 149 (66%) | 40 (18%) | 38 (17%) | 227 (19%) |
| D-dimer (mg/dL)[b] | | | | |
| Normal (≤0·5 µg/mL) | 674 (85%) | 75 (9.4%) | 45 (5.7%) | 794 (67%) |
| Elevated (>0·5 µg/mL) | 282 (73%) | 62 (16%) | 41 (11%) | 385 (33%) |
| Years since HIV diagnosis, median (IQR)[b] | 6.9 (2.8–11) | 3.7 (1.4–8.3) | 6.2 (1.9–11) | 6.6 (2.6–11) |
| Years since starting ART, median (IQR) | 6.6 (2.4–11) | 3.0 (1.0–8.1) | 6.0 (1.8–9.9) | 6.1 (2.1–11) |
| pVL log$_{10}$ (copies/mL), median (IQR) | 1.3 (1.3–1.3) | 2.0 (1.8–2.4) | 4·2 (3.4–4.9) | 1.3 (1.3–2.1) |
| CD4 count (cells/µL), median (IQR) | 604 (411–818) | 430 (280–644) | 313 (192–514) | 558 (362–779) |
| CD4 count (cells/µL)[b] | | | | |
| ≥500 | 579 (89%) | 49 (7.5%) | 22 (3.4%) | 650 (58%) |
| 200–499 | 271 (72%) | 65 (17%) | 39 (10%) | 375 (34%) |
| <200 | 52 (56%) | 18 (19%) | 23 (25%) | 93 (8%) |
| ART regimen at enrollment[d] | | | | |
| Efavirenz (EFV)/Nevirapine (NVP) | 216 (77%) | 26 (9.3%) | 39 (14%) | 281 (23%) |
| Dolutegravir (DTG)-based regimen | 739 (83%) | 108 (12%) | 47 (5.3%) | 894 (74%) |
| Switched from NNRTI- to DTG-based regimen[e] | 361 (84%) | 49 (11%) | 22 (5.1%) | 432 (48%) |
| Started ART on DTG-based regimen | 368 (82%) | 57 (13%) | 24 (5.3%) | 449 (50%) |
| Lopinavir/ritonavir (LPV/r)/Nelfinavir (NFV)[f] | 15 (58%) | 7 (27%) | 4 (15%) | 26 (2.2%) |
| Past ARV history | | | | |
| Zidovudine (AZT) | 204 (84%) | 20 (8.2%) | 19 (7.8%) | 243 (20%) |
| EFV/NVP | 504 (51%) | 58 (9.7%) | 36 (6.0%) | 598 (50%) |
| DTG | 0 | 0 | 1 (100%) | 1 (0.1%) |
| LPV/r/NFV | 16 (70%) | 4 (17%) | 3 (13%) | 23 (1.9%) |

*3TC* lamivudine, *ABC* abacavir, *ART* antiretroviral therapy, *ARV* antiretroviral, *BMI* body mass index, *IQR* interquartile range, *NNRTI* non-nucleoside reverse transcriptase inhibitor, *pVL* plasma viral load, *TDF* tenofovir.

[a]Values in this column are denominators for row statistics.

[b]Missing data: Education (N = 18 infants), BMI (N = 4), Hemoglobin (N = 6), D-dimer (N = 22), Years since HIV diagnosis (N = 12), and CD4 count (N = 83).

[c]Normal hemoglobin (g/dL) for: female: ≥12 (pregnant: ≥11), male: ≥13, and children (0·5–4·9 years: ≥11, 5–11: ≥12, 12–14 years: ≥12).

[d]Combined with 3TC+TDF (1167/1201 [97%]), 3TC+AZT (15/1201 [1.2%]), and 3TC+ABC (16/1201 [1.3%]).

[e]The row total excludes 13 individuals (pVL<40 = 10, pVL 40-999 = 2, and pVL >= 1000 = 1) with past exposure to LPV/r-based regimens.

[f]Includes 3 participants on NFV-based ART.

## Factors associated with virologic failure

Among the 1201 participants, 970 (81%) had pVL<40 copies/mL, 141 (12%) had low-level viremia with pVL 40–999 copies/mL, and 90 (7.5%) had virologic failure with pVL≥1000 copies/mL (Table 1). Similar proportions of participants on DTG- or EFV/NVP-based ART had low-level viremia, as did those who started ART on a DTG-based regimen or switched to DTG-from NNRT-based ART (Table 1). In contrast, significantly fewer DTG recipients (47/894, 5.3%) had virologic failure compared with participants

**Table 2 | Association of sociodemographic and clinical factors with virologic failure in PWH at enrolment in HONOR (N = 1060)[a]**

| Characteristics | Total (N = 1060) n | Total Virologic Failure (N = 90) n (%) | Prevalence ratio (95% CI) Unadjusted | Adjusted[b] |
|---|---|---|---|---|
| **Age, years** | | | | |
| ≥50 | 289 | 10 (3.5) | Ref | - |
| 25–49 | 695 | 65 (9.4) | 2.7 (1.4–5.1) | 2.3 (1.2–4.4) |
| 15–24 | 45 | 10 (22) | 6.4 (2.8–15) | 6.1 (2.7–14) |
| <15 | 31 | 5 (16) | 4.7 (1.7–13) | 4.5 (1.3–16) |
| **Sex assigned at birth** | | | | |
| Male | 286 | 25 (8.7) | - | - |
| Female | 774 | 65 (8.4) | 1.0 (0.6–1.5) | 1.0 (0.6–1.6) |
| **Education[c]** | | | | |
| Vocational-University | 190 | 13 (6.8) | - | - |
| Primary-high school | 641 | 55 (8.6) | 1.3 (0.7–2.2) | 1.0 (0.6–1.8) |
| None | 214 | 19 (8.9) | 1.3 (0.7–2.6) | 1.0 (0.5–2.0) |
| **Hemoglobin (g/dL)[c]** | | | | |
| ≥11 | 868 | 52 (6.0) | - | - |
| <11 | 187 | 38 (20) | 3.4 (2.3–5.0) | 2.3 (1.4–3.5) |
| **D-dimer (mg/dL)[c]** | | | | |
| Normal (≤0·5 µg/mL) | 719 | 45 (6.3) | - | - |
| Elevated (>0·5 µg/mL) | 323 | 41 (13) | 2.0 (1.4–3.0) | 1.4 (1.0–2.1) |
| **Time since starting ART, years** | | | | |
| ≤5 | 505 | 45 (8.9) | - | - |
| >5 | 555 | 45 (8.1) | 0.9 (0.6–1.4) | 1.5 (1.0–2.3) |
| **CD4 count (cells/µL)[c]** | | | | |
| ≥500 | 601 | 22 (3.7) | - | - |
| 200–499 | 310 | 39 (13) | 3.4 (2.1–5.7) | 3.4 (2.0–5.8) |
| <200 | 75 | 23 (31) | 8.4 (4.9–14) | 7.3 (4.0–13) |
| **Current ART use** | | | | |
| **Efavirenz** | | | | |
| No | 815 | 55 (6.7) | - | - |
| Yes | 245 | 35 (14) | 1.9 (1.2–2.9) | 2.8 (1.8–4.3) |
| **Nevirapine** | | | | |
| No | 1050 | 86 (8.2) | - | - |
| Yes | 10 | 4 (60) | 4.9 (2.2–11) | 4.7 (2.0–11) |
| **Tenofovir** | | | | |
| No | 28 | 8 (29) | - | - |
| Yes | 1032 | 82 (7.9) | 0.4 (0.2–0.9) | 0.6 (0.2–1.7) |
| **Zidovudine** | | | | |
| No | 1046 | 83 (7.9) | - | - |
| Yes | 14 | 7 (50) | 5.4 (2.8–10) | 4.3 (1.9–9.8) |
| **Lopinavir/ritonavir** | | | | |
| No | 1044 | 86 (8.2) | - | - |
| Yes | 16 | 4 (25) | 3.1 (1.3–7.4) | 2.1 (0.7–6.3) |
| **Dolutegravir (DTG)** | | | | |
| No | 274 | 43 (16) | - | - |
| Yes | 786 | 47 (6.0) | 0.4 (0.3–0.6) | 0.3 (0.2–0.5) |

**Table 2 (continued) | Association of sociodemographic and clinical factors with virologic failure in PWH at enrolment in HONOR (N = 1060)[a]**

| Characteristics | Total (N = 1060) n | Total Virologic Failure (N = 90) n (%) | Prevalence ratio (95% CI) Unadjusted | Adjusted[b] |
|---|---|---|---|---|
| **DTG-based ART regimen[d]** | | | | |
| Switched from NNRTI- to DTG-based regimen | 383 | 22 (5.7) | - | - |
| Started ART on DTG-based regimen | 392 | 24 (6.1) | 1.1 (0.6–1.9) | 0.7 (0.4–1.3) |
| **InSTI-based vs. NNRTI-based ART** | | | | |
| NNRTI-based ART | 255 | 39 (15) | - | - |
| InSTI-based ART | 786 | 47 (6.0) | 0.4 (0.3–0.6) | 0.3 (0.2–0.5) |

ART antiretroviral therapy, CI confidence interval, InSTI integrase strand transfer inhibitor, LPV/r lopinavir/ritonavir, NNRTI non-nucleoside reverse transcriptase inhibitor, pVL plasma viral load, PWH persons with HIV, VF virologic failure.
[a]We compared individuals with pVL<40 copies/mL (N = 970) to individuals with pVL≥1000 copies/mL (N = 90).
[b]Adjusted for age, sex, education, hemoglobin, D-dimer, time on ART, and CD4 count.
[c]Missing data: Education (N = 15 infants), Hemoglobin (N = 5), D-dimer (N = 18), and CD4 count (N = 74).
[d]Excludes 11 individuals (pVL <40 = 10 and pVL ≥1000 = 1) with past exposure to LPV/r-based regimen.

on an EFV/NVP-based regimen (39/281, 14%, adjusted PR [aPR]=0.3, 95% CI: 0.2–0.5) (Tables 1 and 2). Younger age (compared to age≥50; <15 years: aPR=4.5 [95% CI: 1.3–16]; 15–24 years: aPR=6.1 [95% CI: 2.7–14]; 25–49 years: aPR=2.3 [95% CI: 1.2–4.4]), hemoglobin <11 g/dL (aPR=2.3, [95% CI: 1.4–3.5]) and lower CD4 count (compared to CD4 ≥ 500; CD4 < 200 cells/µL: aPR=7.3 [95% CI: 4.0–13] and 200–499 cells/µL: aPR=3.4 [95% CI: 2.0–5.8]) were significant risk factors for virologic failure (Table 2). The association with sex at birth, education level, D-dimer levels and ART duration did not reach statistical significance. Treatment with EFV, NVP, or zidovudine (AZT) was associated with increased risk, while DTG (aPR=0.3, [95% CI: 0.2–0.5]) was associated with reduced virologic failure risk (Table 2). Among DTG recipients, virologic failure risk was similar between those who started ART on a DTG-based regimen and those who switched from an NNRTI-based to a DTG-based regimen (Table 2). Those on a first line DTG regimen were on ART for a shorter duration (2.5 [IQR 0.6–9.4] years) than those who had switched regimens (8.8 [IQR 4.9–12] years) (Supplementary Table 2). Similar risk factors for virologic failure were identified in a stratified analysis limited to participants on DTG-containing ART (Supplementary Table 3). Stratification of results for children under 15 years of age was precluded by the small sample size.

**Prevalence and correlates of HIV drug resistance**
GDR testing was successful in samples from 70 of the 90 (78%) participants with virologic failure. Thirty-one of 70 samples (11 DTG and 20 non-DTG) were found to have drug resistance mutations called at a frequency of ≥5% but not at a frequency ≥15%. IN Accessory mutations were observed in three DTG and two non-DTG samples, and one non-DTG sample exhibited an IN major mutation at the 5% threshold but not at the 15% level. No differences in PR mutations were observed among DTG patients, but differences in PR accessory mutations were identified in three non-DTG samples. Additionally, differences in major PR mutations were observed in two non-DTG samples when comparing the 5% and 15% thresholds. Differences in NNRTI mutations between these thresholds were found in eight DTG and seven non-DTG samples, while differences in NRTI mutations were found in five DTG and nine non-DTG samples. Among those 70 participants with

## Table 3 | Prevalence of resistance to ARV class and drug resistance mutations in PWH whose HIV was sequenced (N = 70)

| Level of resistance and key DRM by ART class | N (%) |
|---|---|
| No resistance to any of 25 ARVs[a] | 13 (19) |
| Some resistance to at least one ARV in any class[b] | 57 (81) |
| **Low- to high-level resistance: any ARV in drug class** | |
| NRTI | 43 (61) |
| NNRTI | 55 (79) |
| InSTI | 4 (5.7) |
| PI | 5 (7.1) |
| **Any intermediate/high resistance to ≥2 classes** | |
| NRTI + NNRTI | 42 (60) |
| NRTI + InSTI | 3 (4.3) |
| NNRTI + InSTI | 3 (4.3) |
| NNRTI + PI | 4 (5.7) |
| NRTI + PI | 5 (7.1) |
| PI + InSTI | 0 |
| InSTI + NRTI + NNRTI | 3 (4.3) |
| PI + NRTI + NNRTI | 4 (5.7) |
| **Specific major DRM by ARV class** | |
| **NRTI** | |
| M41L | 5 (7.1) |
| K65R | 13 (19) |
| D67N | 6 (8.6) |
| K70E/G/N/Q/R/T | 16 (23) |
| M184I/V | 42 (60) |
| T215F/Y | 9 (13) |
| **NNRTI** | |
| K101E/P | 15 (21) |
| K103H/N/S | 35 (50) |
| Y181C | 15 (21) |
| Y188F/H/L | 6 (8.6) |
| G190A/E/S | 21 (30) |
| P225H | 12 (17) |
| **InSTI** | |
| T66A | 1 (1.4) |
| G118R | 2 (2.9) |
| E138K | 1 (1.4) |
| R263K | 1 (1.4) |
| **PI** | |
| M46I | 3 (3.2) |
| I54T/V | 1 (1.4) |
| L76V | 2 (2.9) |

*3TC* lamivudine, *ART* antiretroviral therapy, *ARV* antiretroviral, *DRM* drug resistance mutation, *EFV* efavirenz, *InSTI* integrase strand transfer inhibitor, *NNRTI* non-nucleoside reverse transcriptase inhibitor, *NRTI* nucleoside reverse transcriptase inhibitor, *NVP* nevirapine, *PI* protease inhibitor, *PWH* persons with HIV, *TDF* tenofovir.
[a]Includes 4 individuals with potential low-level resistance.
[b]Low-, intermediate-, or high-level resistance.

virologic failure and successful GDR testing, 56 (80%) were infected with HIV-1 clade CRF02_AG; 32 (46%) were on DTG-based ART; 34 (49%) were on EFV/NVP-based ART; and 4 (5.7%) were on LPV/r-based ART (Supplementary Table 4). Fifty-seven (81%) had low-, intermediate-, or high-level HIVDR to ≥1 of the 25 ARVs evaluated, while 13 (19%) had no

resistance to any of the ARVs tested (Table 3). Fifty-five (79%) had resistance to at least one NNRTI and 43 (61%) to at least one NRTI. Forty-two (60%) had intermediate- or high-level resistance to both NRTIs and NNRTIs. Resistance to InSTIs and PIs was low, at 5.7% and 7.1%, respectively.

Analysis of estimated resistance to individual NRTIs revealed intermediate- or high-level resistance to TDF 24% and 61% to 3TC/FTC (Table 4). This was due primarily to the high prevalence of the drug resistance mutations K65R (19%), K70E/G/N/Q/R/T (23%), and M184I/V (60%) (Table 3). Intermediate-high resistance to EFV and NVP was found in 74% and 75%, respectively (Table 4), with high prevalence of the drug resistance mutations K103H/N/S (50%), G190A/E/S (30%), K101E/P (21%), and Y181C (21%) (Table 3). Two (2.9%) and three (4.3%) individuals had intermediate-high resistance to DTG and LPV, respectively, which are the only InSTI and PI in standard use in Liberia (Table 4). Major drug resistance mutations to InSTIs included G118R (2.9%), T66A (1.4%), E138K (1.4%), and R263K (1.4%). Major drug resistance mutations to PIs included M46I (3.2%), L76V (2.9%), and I54T/V (1.4%) (Table 3).

We also assessed resistance in the context of current ART regimens. In comparisons between those on non-DTG- versus DTG-based regimens, the prevalence of major NRTI and NNRTI-associated drug resistance mutations was high in both groups but tended to be higher in participants on non-DTG- versus DTG-based regimens for the K65R (26% v. 9%), D67N (16% v. 0%), M184I/V (76% v. 41%), K101E/P (32% v. 9%), and K103H/N/S (66% v. 31%) drug resistance mutations (Fig. 1) (Supplementary Data 1).

Focusing on an analysis of resistance in NNRTI+2NRTIs versus InSTI +2NRTIs regimens, we found that among 34 individuals with virologic failure on an NNRTI+2NRTIs regimen, 100% had intermediate-high resistance to at least one drug in the regimen: 11 (32%) to all three drugs, 16 (47%) to two drugs, and 7 (21%) to one of three drugs in that regimen (Table 5). Of the 32 individuals with virologic failure on an InSTI (DTG) +2NRTIs regimen, none had resistance to InSTIs alone, but 13 (41%) had intermediate-high resistance to at least one drug in the regimen: 2/32 (6.3%) to DTG and both NRTIs, 3/32 (9.4%) to both NRTIs, and 8/32 (25%) to 1 NRTI (Table 5; Supplementary Table 5). Of the two InSTI+2NRTIs recipients with multi-class-resistant HIV, one participant had HIV and ARV exposure since 2008 (Supplementary Table 6). The second individual self-reported recent HIV infection and initiation of DTG-based ART shortly before study enrolment but had evidence of advanced disease with a CD4 count of 15 cells/μL. In those on an InSTI+2NRTIs regimen, the prevalence of intermediate-high HIVDR to any drug was lower in those whose initial regimen was DTG-based (1/15, 7%) versus those who had switched to a DTG-based regimen (12/17, 71%) (aPR=0.1 [95%CI: 0.01–0.8]) (Table 5). The intermediate-high HIVDR observed in both InSTI+2NRTIs and NNRTI+2NRTIs regimens was mainly driven by resistance to the NRTI backbone (HIVDR to one or both NRTIs in 13/32 (41%) versus 27/34 (79%), respectively; aPR=0.5 [95%CI: 0.3–0.8]) (Supplementary Table 5). Participants with HIVDR to the NRTI backbone in their regimen had been on ART for a median 7.9 (IQR 4.0–11) years while those with no resistance to the NRTI backbone were on ART for a shorter duration (median 2.1 [IQR 0.6–8.9] years) (Supplementary Table 5).

## Discussion

In this cohort of PWH in Liberia on ART during the rollout of InSTI-containing regimens, we found a higher prevalence of virus suppression (81%) than observed in a smaller group after the 2014 Ebola outbreak[5]. In our cohort, a lower percentage on a DTG-based regimen had virologic failure compared with those on EFV/NVP-based ART (5.3% v. 14%, respectively), likely reflecting the higher genetic barrier to resistance of this second generation InSTI compared to NNRTIs[14]. Our report, one of the few from West Africa, adds to a growing number suggesting a relatively high prevalence of virus suppression by DTG-containing regimens in sub-Saharan Africa[15–18].

In our cohort, 7.5% of ART recipients had pVL≥1000 copies/mL at their baseline visit. Younger age, lower CD4 count, longer time on ART,

**Table 4 | Prevalence of HIVDR to antiretroviral drugs in PWH with VL ≥ 1000 copies/mL successfully sequenced (N = 70)**

| Antiretroviral drug | Level of resistance | | | | No resistance |
|---|---|---|---|---|---|
| | High | Intermediate | Low | Potential low[a] | |
| **NRTI** | | | | | |
| Tenofovir | 4 (5.7%) | 13 (19%) | 12 (17%) | 1 (1.4%) | 40 (57%) |
| Lamivudine/Emtricitabine | 42 (60%) | 1 (1.4%) | 0 | 0 | 27 (39%) |
| Abacavir | 18 (26%) | 12 (17%) | 13 (19%) | 0 | 27 (39%) |
| Didanosine | 21 (30%) | 8 (11%) | 5 (7.1%) | 9 (13%) | 27 (39%) |
| Stavudine | 13 (19%) | 14 (20%) | 7 (10%) | 0 | 36 (51%) |
| Zidovudine | 11 (16%) | 1 (1.4%) | 2 (2.9%) | 2 (2.9%) | 54 (77%) |
| **NNRTI** | | | | | |
| Efavirenz | 51 (73%) | 1 (1.4%) | 1 (1.4%) | 2 (2.9%) | 15 (21%) |
| Nevirapine | 52 (74%) | 1 (1.4%) | 2 (2.9%) | 2 (2.9%) | 13 (19%) |
| Doravirine | 17 (24%) | 20 (29%) | 4 (5.7%) | 8 (11%) | 21 (30%) |
| Etravirine | 15 (21%) | 13 (19%) | 7 (10%) | 13 (19%) | 22 (31%) |
| Rilpivirine | 31 (44%) | 5 (7.1%) | 8 (11%) | 4 (5.7%) | 22 (31%) |
| **InSTI** | | | | | |
| Dolutegravir | 2 (2.9%) | 0 | 0 | 1 (1.4%) | 67 (96%) |
| Bictegravir | 1 (1.4%) | 1 (1.4%) | 0 | 1 (1.4%) | 67 (96%) |
| Cabotegravir | 2 (2.9%) | 0 | 0 | 1 (1.4%) | 67 (96%) |
| Elvitegravir | 3 (4.3%) | 0 | 1 (1.4%) | 9 (13%) | 57 (81%) |
| Raltegravir | 2 (2.9%) | 0 | 1 (1.4%) | 9 (13%) | 58 (83%) |
| **PI** | | | | | |
| Lopinavir | 2 (2.9%) | 1 (1.4%) | 1 (1.4%) | 0 | 66 (94%) |
| Nelfinavir | 5 (7.1%) | 0 | 0 | 0 | 65 (93%) |
| Atazanavir | 3 (4.3%) | 0 | 2 (2.9%) | 0 | 65 (93%) |
| Darunavir | 0 | 1 (1.4%) | 1 (1.4%) | 1 (1.4%) | 67 (96%) |
| Fosamprenavir | 3 (4.3%) | 0 | 0 | 1 (1.4%) | 66 (94%) |
| Indinavir | 2 (2.9%) | 1 (1.4%) | 2 (2.9%) | 0 | 65 (93%) |
| Saquinavir | 2 (2.9%) | 0 | 3 (4.3%) | 0 | 65 (93%) |
| Tipranavir | 0 | 2 (2.9%) | 1 (1.4%) | 0 | 67 (96%) |

*HIVDR* HIV drug resistance, *InSTI* integrase strand transfer inhibitor, *NNRTI* non-nucleoside reverse transcriptase inhibitor, *NRTI* nucleoside reverse transcriptase inhibitor, *PI* protease inhibitor, *PWH* persons with HIV, *VL* viral load.

[a]Potential low-level resistance is included here for completeness but excluded from other analyses. The Stanford HIV Drug Resistance Database defines potential low-level resistance as viruses with sequences containing mutations that may indicate previous ARV exposure or that may contain mutations that are associated with drug resistance only when they occur with additional mutations.

lower hemoglobin, and currently taking EFV, NVP, or AZT were associated with higher virologic failure risk. Lower CD4 count and longer ART duration were associated with virologic failure in a recent review of PWH in sub-Saharan Africa, which also identified sub-optimal adherence and tuberculosis co-infection as risk factors[19]. The association between younger age and virologic failure has been attributed to various factors, including reduced adherence and HIVDR related to significant ART experience in perinatally infected adolescents[20,21]. Higher virologic failure risk in our participants with lower hemoglobin levels was consistent with the finding that anemia independently predicted increased odds of virologic failure at 72 weeks of ART in a Nigerian cohort[22], and published associations between anemia and poor clinical outcomes, including HIV disease progression and increased mortality[23,24]. In settings with limited access to pVL monitoring and GDR testing, these routinely measured variables might be useful as initial indicators of virologic failure to prompt closer monitoring and earlier assessment.

HIVDR was highly prevalent in those with virologic failure whose virus was successfully sequenced, with low- to high-level resistance to at least one ARV in 81%. This was mainly composed of resistance to NNRTIs (79%) and NRTIs (61%) or to both NRTIs and NNRTIs (60%). This was similar to the prevalence of drug resistance mutations in those receiving NNRTI-based ART in the only previous study of HIVDR in Liberia[5], and in adults and

children on ART from other countries in sub-Saharan Africa[25–30]. Notably, our study is among the few in sub-Saharan Africa, and the largest to our knowledge in West Africa, to extend previous observations by also investigating resistance to InSTIs and PIs[25,29–31]. In our participants with virologic failure, the prevalence of any resistance to InSTIs and PIs was low (5.7% and 7.1%, respectively), with much lower rates of resistance to DTG (2.9%) and LPV (5.7%), consistent with recent reports[25,29,30].

The prevalence of intermediate-high resistance to the NRTI backbone that the participants were currently taking was high. All 34 individuals (100%) with virologic failure on an NNRTI-based regimen had intermediate-high resistance to at least one drug in the regimen, and 27/34 (79%) had resistance to one or both NRTIs. A lower percentage of individuals with virologic failure on a DTG-based regimen had resistance (13/32, 41%), but all 13 had intermediate-high resistance to at least one drug in the NRTI backbone.

The prevalence of DTG resistance among participants with virologic failure on a DTG-containing regimen whose virus was successfully sequenced was low (2/32, 6.3%). These two InSTI+2NRTIs recipients had multi-class-resistant HIV with InSTI drug resistance mutations known to be selected by DTG and to impact virus susceptibility to DTG, including G118R, T66A, E138K, and R263K. Although one of these participants may represent a rare case of transmitted InSTI resistance, it is more likely that he

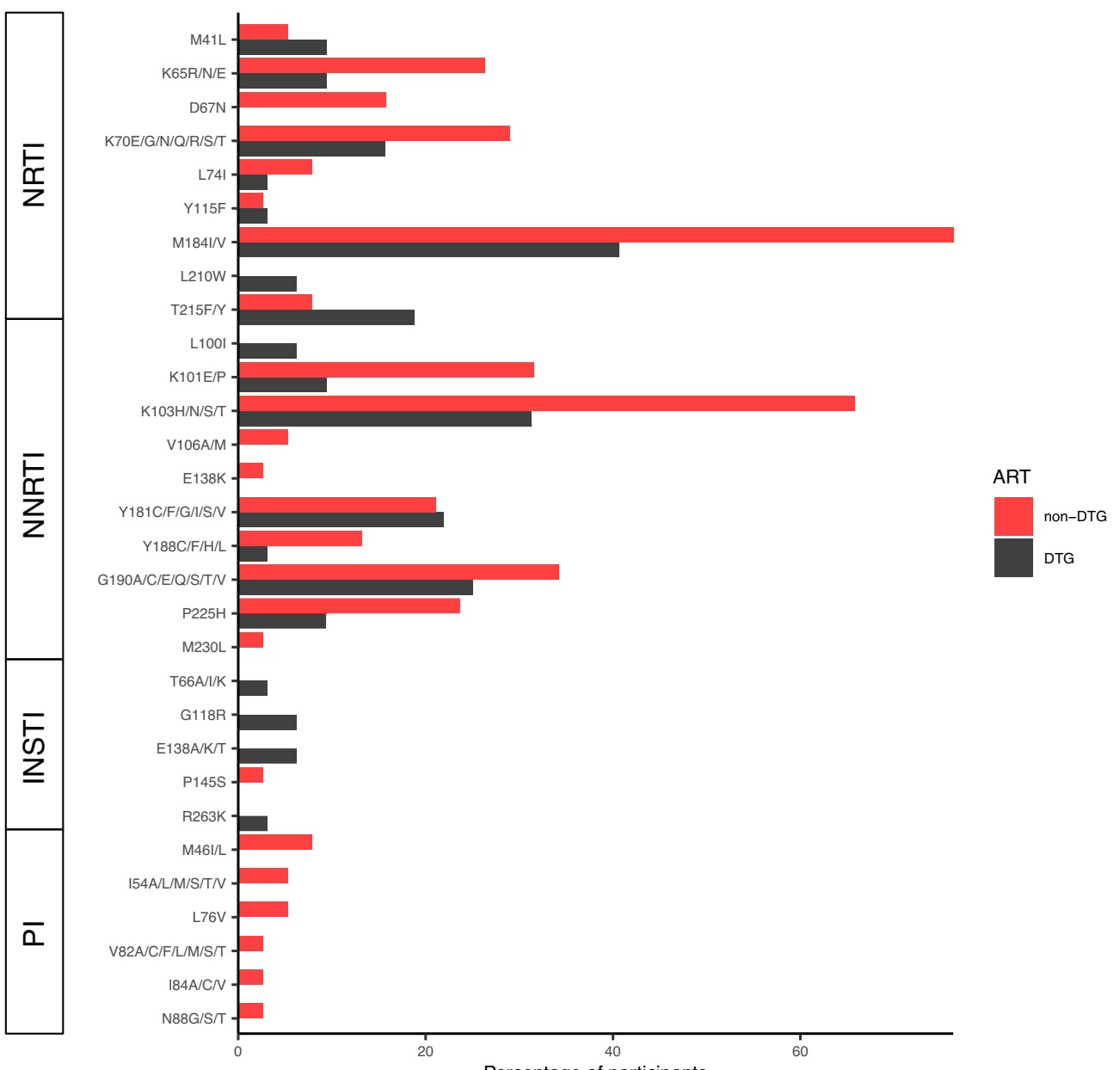

**Fig. 1 | Major HIV mutational sequences in PWH with pVL pVL≥1000 copies/mL (*N* = 70).** The percentages of participants (x-axis) with HIV-1 sequences containing major drug resistance mutations (y-axis) are shown for those receiving non-dolutegravir (DTG)-based (red bars, *N* = 38) and DTG-based (black bars, *N* = 32) antiretroviral therapy. Not all listed mutations at a given position were detected. NRTI mutations not detected: S68del, T69ins/del, 0151L/M. NNRTI mutations not detected: F227C. INSTI mutations not detected: E92G/O/V, F121Y, G140A/C/R/S, Y143A/C/G/H/K/R/S, 0146P, S147G, 0148H/K/N/R, V151A/L, and N155H/S/T. PI mutations not detected: D3ON, V32I, 147A/V, G48A/L/M/O/S/T/V, I5OL/V, and L90M.

did not disclose prior ARV exposure that had selected for resistance, as suggested by the detection of the NRTI and NNRTI drug resistance mutations. Thus, in our cohort, for those who had virologic failure while on DTG-based ART, NRTI resistance probably predated DTG exposure resulting in treatment with DTG + ≤1 NRTI. DTG functional monotherapy has been associated with faster progression to virologic failure, accumulation of more drug resistance mutations, and poor treatment outcome[31].

In aggregate, our data support the evidence of other studies in sub-Saharan Africa regarding the benefit of fast-tracking the transition from NNRTI-based to DTG-based regimens[2]. Our results also suggest there might be value in performing HIVDR testing in persistently viremic patients receiving failing regimens to inform salvage ARV selection. Even though InSTI-related drug resistance mutations were detected in only 2 of 32 participants who failed a DTG-based regimen in our study, in light of

current widespread use of DTG in Liberia, ongoing population-level surveillance for DTG resistance over time will be important to inform future policy on when to perform HIVDR testing at an individual level. The NADIA randomized controlled trial conducted in Africa showed that DTG plus two NRTIs produced durable suppression at 96 weeks, including among subjects with pre-existing NRTI resistance, but DTG was at greater risk of resistance than darunavir in second-line therapy[32]. Additional studies are needed to assess the ability of InSTI-based regimens to maintain longer-term virus suppression, particularly in those with NRTI- and NNRTI-related drug resistance mutations[15,33], in sub-Saharan Africa.

Although only 7.5% of participants in our study had pVL≥1000 copies/mL, 12% had low-level viremia (pVL 40–999 copies/mL), including 108 DTG recipients. These individuals may be at higher risk of progression to virologic failure and accumulation of drug resistance mutations compared

**Table 5 | HIV drug resistance to current ART regimen in PWH with pVL≥1000 copies/mL successfully sequenced (*N* = 70)**

| Current ART regimen | N | Interm/high resistance, all drugs in regimen | Interm/high resistance, two drugs in regimen[a] | Interm/high resistance to one drug in regimen[b] | Any Low-to-high Resistance[c] | Susceptible to all ARVs in regimen |
|---|---|---|---|---|---|---|
| NNRTI + 2NRTIs | 34 | 11 (32%) | 16 (47%) | 7 (21%) | 34 (100%) | 0 |
| DTG + 2NRTIs[d] | 32 | 2 (6.3%) | 3 (9.4%) | 8 (25%) | 13 (41%) | 19 (59%) |
| Started on DTG-ART | 15 | 1 (6.7%) | 0 | 0 | 1 (7%) | 14 (93%) |
| NNRTI to DTG-ART | 17 | 1 (5.9%) | 3 (18%) | 8 (47%) | 12 (71%) | 5 (29%) |
| PI + 2NRTIs | 4 | 1 (25%) | 2 (50%) | 1 (25%) | 4 (100%) | 0 |
| Overall, N (%) | 70 | 14 (20%) | 21 (30%) | 16 (23%) | 51 (73%) | 19 (27%) |

*ART* antiretroviral therapy, *ARV* antiretroviral, *DTG* dolutegravir, *EFV* efavirenz, *HIVDR* HIV drug resistance, *InSTI* integrase strand transfer inhibitor, *LPV* lopinavir, *NNRTI* non-nucleoside reverse transcriptase inhibitor, *NRTI* nucleoside reverse transcriptase inhibitor, *NVP* nevirapine, *PI* protease inhibitor, *PR* prevalence ratio, *PWH* persons with HIV, *pVL* plasma viral load.
[a]Intermediate/high resistance to 2 (either both NRTIs or 1 NRTI & base [DTG/LPV/EFV/NVP]), and no HIVDR to third ARV in regimen.
[b]Intermediate/high-level resistance to 1 ARV in regimen (with either low-level resistance to remaining one or no HIVDR to remaining two ARVs in regimen).
[c]Values in this column represent totals of row entries from the preceding three columns (total N with intermediate/high resistance to at least one drug in the regimen); these values plus entries in the last column (N susceptible to all ARVs in regimen) equal 100% of those on a current ART regimen.
[d]Among persons on DTG-based ART, the risk of HIVDR to the NRTI backbone, adjusted for time on ART, was lower in those who started on DTG-ART compared to those who switched from NNRTI-ART to DTG-ART (unadjusted PR = 0.1 [95%CI: 0.01–0.6]; adjusted PR = 0.1 [95%CI: 0.01–0.8]).

with fully suppressed individuals[34]. Longitudinal analyses will be needed to determine whether individuals with low-level viremia at enrolment in our study represent cases of persistent low-level viremia, isolated viral blips, early virologic failure, or declining pVL levels after ART initiation or switching proximate to their baseline visit.

We note some important limitations in this study. Sequencing was unsuccessful in 20 (22%) individuals with virologic failure (most likely due to lower pVL levels and/or mismatches between primer and targets sequences), although this failure ratio is comparable to other recent reports[15,26]. Secondly, medical records and ARV treatment cards were not consistently available to confirm self-reported ART history. Thirdly, we did not sequence samples from individuals with low-level viremia and do not know if they harbored major HIVDR. Fourthly, we enrolled a heterogeneous population on ART motivated to return for serial study visits who may not be representative of all PWH on ART in Liberia. Lastly, we did not collect medication adherence information during the baseline visit, and therefore could not assess its relationship to virologic failure.

Notwithstanding these limitations, our findings have important implications for the care of Liberian PWH. They provide support for starting PWH on a DTG-based regimen and switching those on NNRTI-based ART to regimens containing DTG, followed by support to ensure optimal adherence. In clinical trials, the majority of patients achieved viral suppression by 12 weeks after starting a DTG-based regimen[35]. Inability to suppress pVL after three to four months should prompt evaluation for possible drug resistance and other causes of virologic failure, including barriers to adherence, significant drug-drug interactions causing subtherapeutic ARV concentrations, or impaired intestinal absorption[6]. However, current Liberian HIV guidelines recommend pVL testing at six months after initiating or switching ART, which may delay identifying patients with suboptimal treatment response. Routine GDR testing is not currently available in Liberia, but if available, could identify HIVDR in those experiencing virologic failure for early intervention to reduce further DRM accumulation, preserve future treatment options, and improve clinical outcomes. Longer-term pVL surveillance will be important in individuals on DTG-based regimens. As a LMIC with limited options for individuals with InSTI virologic failure and resistance, conducting clinical trials to identify optimal salvage regimens will be critical to inform future treatment guidelines.

In conclusion, we found that 81% of ART recipients were virologically suppressed in the HONOR cohort at enrolment during the rollout of DTG-containing regimens. Higher risk of virologic failure was associated with younger age, lower CD4 count, longer time on ART, lower hemoglobin, and currently taking EFV, NVP, or AZT. There was a high prevalence of resistance to NRTIs and NNRTIs in those with

virologic failure whose virus was successfully sequenced, but rare resistance to InSTIs and PIs. Despite improvements in HIV-related services for Liberian PWH, significant work remains if Liberia is to achieve the 95-95-95 UNAIDS targets by 2030[8].

## Data availability

All data for the study are not openly available because the parent cohort study is still ongoing but the data can be accessed from the corresponding author on reasonable request. The source data for Fig. 1 is in Supplementary Data 1.

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

## Acknowledgements

This project has been funded in whole or in part with federal funds by the Division of Intramural Research, NIAID, NIH (Contract No. HHSN261200800001E), the Fogarty International Center, NIH (U2R TW011281), and the National Cancer Institute, NIH (Prime Contract No. 75N91019D00024). We also wish to acknowledge support from the University of California, San Francisco International Traineeships in AIDS Prevention Studies (ITAPS), U.S. NIMH, R25MH123256, and Training in Clinical and Epidemiological Research for Liberia (TRACER), U2RTW011281. The study sponsors had no role in the design, data collection, analysis, or writing of this manuscript. The content of this publication does not necessarily reflect the views or policies of the Department of Health and Human Services, nor does mention of trade names, commercial products or organizations imply endorsement by the U.S. Government. We thank all the participants in the HONOR study and collaborating HIV clinics. We also acknowledge the contributions of the following individuals for their assistance with data collection (John Bruce Tweh, Emmanuel Rogers, Korlia Bornawolo, Sarah J. Holder, Christina Andrews, Paul E.K. Bee, Augustus Wallace, Miriam Falika, Princess Lobbo, Joseph Cooper, and Victor D. Taryor), input in study design (Elizabeth Higgs, Moses Badio) and study management (James T. Duworko, Mary Smolskis, Laura McNay, Wissedi Njoh, Melvin Johnson, Kokulo Franklin, Sianneh Tamba, Jestina Doe-Anderson, Leslie Nielsen, and Jemee Tegli).

## Author contributions

J.S.M., S.A.M., A.K.P., C.R., I.W., J.M., C.J.N., S.K., and K.L.J. conceptualized the study. G.G. and C.R. curated and managed the data. J.S.M., I.W., S.A.M., A.K.P., C.R., G.G., J.N.M., C.L., and D.G. devised the methodology. J.S.M., G.G., and C.R., conducted the formal analysis and validation. C.R. created the figures. S.A.M., A.K.P., C.R., and R.D. contributed resources. J.S.M., A.K.P., S.K., G.G., C.R., J.D., O.T.B., E.A., B.W., R.S., A.L.P., S.L.K., T.M., K.L.J., I.W., and S.A.M. provided project administration and supervision. J.S.M. wrote the

original draft of the manuscript. J.S.M., J.M., R.D., G.G., J.M., C.R., A.K.P., S.K., I.W., and S.A.M. directly accessed and verified the underlying data reported in the manuscript. J.D., O.T.B., R.S., C.J.N., K.L.J., J.S.M., I.W., M.M., J.M., B.T.S., W.C., L.D., M.A.K., H.H., R.D., C.R., G.G., and I.M. conducted investigations, data collection, and validation. J.S.M., G.G., C.R., I.W., S.K., A.K.P., and S.A.M. had full access to all the data in the study. All authors reviewed and edited the manuscript and had final responsibility for the decision to submit for publication.

## Competing interests

The authors declare no competing interests.

## Additional information

[1]Partnership for Research on Vaccines & Infectious Diseases in Liberia (PREVAIL), Monrovia, Liberia. [2]University of California San Francisco, San Francisco, CA, USA. [3]National Institute of Allergy and Infectious Diseases (NIAID), National Institutes of Health (NIH), Bethesda, MD, USA. [4]Clinical Research Directorate, Frederick National Laboratory for Cancer Research, Frederick, MD, USA. [5]Division of Biostatistics, School of Public health, University of Minnesota, Minneapolis, MN, USA. [6]Frederick National Laboratory for Cancer Research, Frederick, MD, USA. [7]Advanced BioMedical Laboratories, LLC, Cinnaminson, NJ, USA. [8]Clinical Monitoring Research Program Directorate (CMRPD), Frederick National Laboratory for Cancer Research, Frederick, MD, USA. [9]John F. Kennedy Medical Center, Monrovia, Liberia. [10]Axle Informatics, North Bethesda, MD, USA.
✉e-mail: SMoses@prevailcr.org

