## [Transparent Peer Review file · Communications Medicine]

HIV-1 suppression and rare dolutegravir resistance in antiretroviral-experienced people with HIV in Liberia

Corresponding Author: Dr James Soka Moses

Version 0:

Reviewer comments:

Reviewer #1

(Remarks to the Author)

Summary: This is a cross-sectional study amongst a heterogeneous group of people living with HIV in Liberia. The authors present an analysis of >1200 PWH who are on ART and have viral load data available. Data on virologic failure and drug resistance are presented for people who had been on a wide range of regimens including NNRTIs, INSTIs, and PIs. Most people were on DTG-containing regimens given that the study was done during the transition to TLD. The prevalence of virologic failure, defined as a VL >1,000 copies/mL, among people on DTG was 5.3%. Among 32 people with virologic failure on DTG who had successful viral loads, DTG resistance was only identified in 2 people. This study is unique in that it was conducted in West Africa, a region often under-represented in the literature. Findings are a relevant contribution to the growing amount of international data on HIVDR among people failing DTG-containing regimens.

Major Comments:

1. The paper gives much focus to resistance to NNRTIs, which is much less relevant in the TLD era. The paper's relevance could be improved by placing greater focus on those who failed DTG-containing regimens, given that virologic failure and HIVDR prevalence estimates have been thoroughly described in the literature for NNRTIs. In addition, the regression models examine factors associated with virologic failure and looks at regimen as a predictor of interest. However, in the current era, most individuals are on DTG, so the relevance of findings could be improved by presenting a stratified analysis looking at factors associated with VF on DTG-containing regimens.
2. Methods: The inclusion criteria for the HONOR study are not presented. Therefore, it is difficult to understand from whom the sample was derived. This can be inferred but is not clear until reading Table 1. For example, it would be important to clearly state that children (including infants) are included. If children, infants, and adolescents are included, it would be important to stratify results for these groups as well.
3. Methods: In the regression model, it is not clear how missing data were accounted for in the adjusted models. For example, were missing values imputed, or was complete case analysis employed such that anyone with any missing variable in the multivariable model was excluded?
4. Discussion: The authors state that results suggest that HIVDR testing would be valuable (line 207). However, the evidence supporting this conclusion is not clearly explained, particularly with a very low prevalence of DTG resistance in this study. This could be part of the discussion. However, I didn't agree that this strategy was supported by the study results.
5. Finally, it is challenging to follow the denominators throughout the manuscript. While n and % are frequently provided, the denominator is often not listed, and it is then challenging to correctly interpret the prevalence estimate. I had difficulty finding a clear prevalence statement regarding the prevalence of DTG resistance among those with VF on a DTG-containing regimen, for example, which is a critically important estimate to highlight.

Minor Comments:

1. Background: Line 67, statement should be qualified to say that "little is known in Liberia about the proportion with HIVDR.
2. Background: Line 68, the 95-95-95 targets should be attributed to UNAIDS, rather than WHO.
3. Methods: The specific HIV drug resistance assay used should be stated (i.e. which NGS assay?)
4. Methods: Given that NGS was performed, it is important to state at which threshold resistance mutations are reported. Typically, the threshold often used for clinical decision making is when mutations occur at the 20% threshold. If a lower threshold was used here, that would be critical to state.
5. Results: Though 1276 participants were enrolled, only 1201 had a viral load. It is not clear why 75 people did not undergo VL testing. Similarly, 20 people had failed genotypes. It would be helpful to understand the reasons for this. In addition, it would be helpful to understand any bias generating by missing data at these steps. Supplemental appendices could include data showing any differences between those who did not have VLs and GRTs as compared to those who did.
6. Discussion: Limitations section should further emphasize the limitation of self-reported data, particularly if participants were asked to self-report current regimen and regimen history data. Data have shown that many people do not disclose prior

ART use (as high as 30% in some settings) and ability to recall or know regimen history can be very limited.
7. The discussion discusses possible ramifications of dual NRTI resistance for people on DTG-containing regimens. It would be important to discuss and cite the data from NADIA as part of this dialogue as well.

Reviewer #2

(Remarks to the Author)

This is a well written manuscript on a very relevant topic. The scarcity of prior data on virological outcomes and resistance profiles from West-Africa make this an extremely valuable contribution to the field.

I would strongly recommend publication of this work, after addressing some minor comments:

1. Please include a definition of virological failure. The standard definition is VL>1000 copies/mL on two consecutive occasions, but it looks like a single VL>1000 copies/mL is used in this manuscript.
2. Line 98: please explain $\geq 5\%$ with potential ambiguity. I assume any drug resistance mutation present at $\geq 5\%$ was included on the analysis. It would be interesting to know how many mutations were present $>20\%$ (Sanger consensus) and 5-20%
3. The 20% failure rate for genotyping is mentioned as a limitation; was this random or linked to pVL, collection period, collection site,...?
4. Line 207: it is unclear to which guidelines this is referring to (ref 27), are those the Liberia or US guidelines?
5. Line 210: was genotyping attempted for any of the samples with VL 40-999 copies/mL?
6. Line 210: was low-level viraemia more common in the INSTI-group compared to the PI and NNRTI-group? Also it would be interesting to know if LLV is more common in PWH who initiated DTG compared to those who switched to DTG from a previous regimen.
7. There has been a lot of discussion on the importance of the denominator when reporting DTG resistance. Since you have a nice cohort with the number of people on DTG treatment known, it would be useful to add the prevalence of DTG resistance among all patients in DTG (suppressed and not suppressed) in addition to the currently reported DTG resistance prevalence among PWH and non-suppression.
8. Table 1: Were some patients in NFV?
9. Table 4: I would suggest to only include ARVs that are currently in use

Reviewer #3

(Remarks to the Author)

This is a very good work. It touches on a very important area. I have reviewed the manuscript and I am attaching a tracking version of the manuscript. The data analysis used is sound and valid and the reproducibility of the current work is very likely.

Version 1:

Reviewer comments:

Reviewer #1

(Remarks to the Author)

The authors have submitted a revised manuscript with updates that strengthen the paper substantially. They have fully addressed the reviewer comments, and I agree with the approaches taken by the authorship team. This paper addresses a timely topic and presents data from a region that is commonly under-represented in the literature.

My only remaining comment is related to the use of a 5% frequency threshold to describe drug resistance mutations. Because this is different than the 20% threshold used in Sanger sequencing and for clinical decision making, further discussion of this in the manuscript is warranted, as it does change the interpretation of the study results and leads to a higher reported prevalence of drug resistance mutations. This was nicely discussed in the response the review letter, and I feel that the text included in the letter should also be included in the manuscript.

Reviewer #2

(Remarks to the Author)

The authors have adequately addressed comments made by all three reviewers. The changes made, have improved the quality of the manuscript.

I have no further comments that should be addressed prior to publication.

Reviewer #3

(Remarks to the Author)

All concerns that were raised have been corrected. The manuscript in the current state can be accepted for publication

Point-by-Point Responses to Referee Comments

Reviewer #1 (Remarks to the Author):

Summary: This is a cross-sectional study amongst a heterogenous group of people living with HIV in Liberia. The authors present an analysis of >1200 PWH who are on ART and have viral load data available. Data on virologic failure and drug resistance are presented for people who had been on a wide range of regimens including NNRTIs, INSTIs, and PIs. Most people were on DTG-containing regimens given that the study was done during the transition to TLD. The prevalence of virologic failure, defined as a VL >1,000 copies/mL, among people on DTG was 5.3%. Among 32 people with virologic failure on DTG who had successful viral loads, DTG resistance was only identified in 2 people. This study is unique in that it was conducted in West Africa, a region often under-represented in the literature. Findings are a relevant contribution to the growing amount of international data on HIVDR among people failing DTG-containing regimens.

Major Comments:

1. The paper gives much focus to resistance to NNRTIs, which is much less relevant in the TLD era. The paper's relevance could be improved by placing greater focus on those who failed DTG-containing regimens, given that virologic failure and HIVDR prevalence estimates have been thoroughly described in the literature for NNRTIs. In addition, the regression models examine factors associated with virologic failure and looks at regimen as a predictor of interest. However, in the current era, most individuals are on DTG, so the relevance of findings could be improved by presenting a stratified analysis looking at factors associated with VF on DTG-containing regimens.

Response: We thank Reviewer #1 for their careful review of our manuscript. At baseline, almost ¼ of our participants were on an NNRTI-based regimen. A greater proportion of them experienced VF than the participants on a DTG-containing regimen. We wanted to document that finding and report on all participants in the analysis (derived from the same population of Liberian PWH) for increased statistical power to assess factors associated with VF. To address the Reviewer's concern, we made some changes in the abstract and throughout the text to prioritize the results of those who failed DTG-containing regimens over those who failed NNRT-containing ones (one example, page 2, lines 41-42). In addition, as the Reviewer suggested, we performed a stratified analysis to examine factors associated with VF restricted to participants on DTG-based regimens. The associated factors for VF in DTG recipients were similar to those found in the larger cohort, which are now included in the text (page 5, lines 154-155) and Supplementary Table 3.

2. Methods: The inclusion criteria for the HONOR study are not presented. Therefore, it is difficult to understand from whom the sample was derived. This can be inferred but is not clear until reading Table 1. For example, it would be important to clearly state that children (including infants) are included. If children, infants, and adolescents are included, it would be important to stratify results for these groups as well.

Response: We thank the reviewer for this valuable suggestion. We have now added the inclusion criteria from our protocol to the Methods (page 3, lines 88-94). While we do show pVL results for 37 children under 15 years old and 54 participants aged 15-24 in Table 1 along with the association between age and virologic failure in Table 2, small sample sizes unfortunately precluded rigorous analyses of results stratified for these young participants. This justification has been added to the Results (page 5, lines 155-156).

3. Methods: In the regression model, it is not clear how missing data were accounted for in the adjusted models. For example, were missing values imputed, or was complete case analysis employed such that anyone with any missing variable in the multivariable model was excluded?

Response: We thank Reviewer #1 for making this important point. We have now clarified in the Methods that due to small numbers of missing values, a complete case analysis was conducted (page 4, lines 123-124). As footnoted in Table 2, analyses were adjusted for age, sex, education level, hemoglobin, d-dimer, time on ART and CD4 count. Missing values for these factors are included in a footnote, as is exclusion of 13 individuals with past exposure to LPV/r-based regimens. Thus, the number of participants in the adjusted analysis is slightly less than in the unadjusted analysis.

4. Discussion: The authors state that results suggest that HIVDR testing would be valuable (line 207). However, the evidence supporting this conclusion is not clearly explained, particularly with a very low prevalence of DTG resistance in this study. This could be part of the discussion. However, I didn't agree that this strategy was supported by the study results.

Response: We agree with Reviewer #1 that our data do not justify resistance testing for every patient who fails a DTG-based regimen at this time. However, as reported in other countries in Africa, there has been an increase in the prevalence of DTG resistance over time paralleling increased prescribing of this ARV drug across the continent. We think that continued surveillance of DTG resistance over time at a population level is important to inform future country-level policy on when to perform resistance testing at an individual level. It also stands to reason that HIVDR testing in persistently viremic patients, despite support to ensure adherence and evaluation for other causes of VF, might be of value, given the availability of alternative ARV options (e.g., protease inhibitors), to inform ARV selection. Revisions reflecting these points were made in the Discussion to define the basis more clearly for this conclusion (page 7, lines 240-245).

5. Finally, it is challenging to follow the denominators throughout the manuscript. While n and % are frequently provided, the denominator is often not listed, and it is then challenging to correctly interpret the prevalence estimate. I had difficulty finding a clear prevalence statement regarding the prevalence of DTG resistance among those with VF on a DTG-containing regimen, for example, which is a critically important estimate to highlight.

Response: Related to this point, we considered adding denominators to each cell

throughout the tables, but this led to overcrowding and loss of clarity. Alternatively, to make the denominators more readily accessible, we made the following revisions to explain their source. Footnotes/Superscripts were used to designate the denominators for row statistics in Tables 1 and 5 and Supplementary Tables 1, 2, and 5. We also made some slight modifications to the formatting of Tables 2 and 5 to enhance clarity. (Tables 3 and 4 and Supplementary Table 4 present data for the 70 participants with VF whose virus was successfully sequenced, which was already highlighted in the Table title.) In addition to these revisions, to specifically address the Reviewer's concern about not finding a clear statement regarding the critically important estimate of the prevalence of DTG resistance among those with VF on a DTG-containing regimen, we made some small revisions in the Results and Discussion to highlight this number (page 5, lines 182, 184-187; page 6, lines 228-229).

Minor Comments:

1. Background: Line 67, statement should be qualified to say that "little is known in Liberia about the proportion with HIVDR. **Reply: Done (page 3, line 73).**

2. Background: Line 68, the 95-95-95 targets should be attributed to UNAIDS, rather than WHO. **Reply: Thank you; done (page 3, line 75).**

3. Methods: The specific HIV drug resistance assay used should be stated (i.e. which NGS assay?)

Response: We have revised this section in the Methods to specify that MiSeq was used (page 4, line 108).

4. Methods: Given that NGS was performed, it is important to state at which threshold resistance mutations are reported. Typically, the threshold often used for clinical decision making is when mutations occur at the 20% threshold. If a lower threshold was used here, that would be critical to state.

Response: We have slightly revised the Methods to emphasize the 5% frequency threshold that was used (page 4, line 109).

5. Results: Though 1276 participants were enrolled, only 1201 had a viral load. It is not clear why 75 people did not undergo VL testing. Similarly, 20 people had failed genotypes. It would be helpful to understand the reasons for this. In addition, it would be helpful to understand any bias generating by missing data at these steps. Supplemental appendices could include data showing any differences between those who did not have VLs and GRTs as compared to those who did.

Response: We agree with Reviewer #1 that this information related to missing data might improve understanding. Seventy-five participants lacked pVL results due to unsuccessful phlebotomy (n=1) or delays related to batch processing of frozen plasma specimens (n=74). Missing pVL data was not attributable to conditions related to collection timing, collection site, specimen storage, pVL assay run, or data analysis. A comparison of baseline characteristics between participants with pVL results and the 75 without pVL results revealed no major

differences between these groups. Appropriate revisions conveying this information were made to the Results (page 4, lines 129-133 and Supplementary Table 1).

Our rate of successful genotyping was comparable to other recent reports, as mentioned in the Discussion. Unsuccessful genotyping in 20 participants was likely due to lower pVL levels and/or mismatches between the primer and target sequences. There was no evidence to suggest that failed attempts occurred due to differences in the way specimens were obtained (timing, site), initially processed, stored, shipped or handled downstream in the sequencing process. Likely reasons for failed genotyping have been added to the Discussion (pages 7, lines 257-258).

6. Discussion: Limitations section should further emphasize the limitation of self-reported data, particularly if participants were asked to self-report current regimen and regimen history data. Data have shown that many people do not disclose prior ART use (as high as 30% in some settings) and ability to recall or know regimen history can be very limited.

Response: We thank Reviewer #1 for sharing this concern. We believe that a unique strength of our study is that most PWH in Liberia possess a “yellow card” containing their current and prior ART regimens, which they present at appointments to study team clinicians. This information is reviewed and used to complete the study case report forms. While better than relying solely on self-reporting, we agree that this is still a potential limitation, as already addressed in the Discussion (page 7, lines 258-259), since not every participant presented their yellow card at their baseline visit.

7. The discussion discusses possible ramifications of dual NRTI resistance for people on DTG-containing regimens. It would be important to discuss and cite the data from NADIA as part of this dialogue as well.

Response: We thank the reviewer for this suggestion. We have added a statement about the important findings in NADIA of pVL suppression to 96 weeks by a DTG-based regimen, including those with pre-existing NRTI resistance, that was non-inferior to darunavir-based regimens, although DTG was at greater risk of resistance than darunavir in second-line regimens (page 7, lines 245-247). This appears to support the importance of our assertion that additional studies are needed to assess the ability of InSTI-based regimens to maintain longer-term virus suppression, particularly in individuals with pre-existing NRTI DRMs.

Reviewer #2 (Remarks to the Author):

This is a well written manuscript on a very relevant topic. The scarcity of prior data on virological outcomes and resistance profiles from West-Africa make this an extremely valuable contribution to the field.

I would strongly recommend publication of this work, after addressing some minor comments:

1. Please include a definition of virological failure. The standard definition is VL>1000 copies/mL on two consecutive occasions, but it looks like a single VL>1000 copies/mL is used in this manuscript.

Response: We thank Reviewer #2 for their careful review of our manuscript and for making this important point. Since this is a cross-sectional analysis of data obtained from the baseline visit of participants in our cohort, we defined virologic failure as a single HIV-1 pVL result of ≥ 1000 copies/mL. Accordingly, we have made appropriate revisions in the Methods to clarify the definition used herein (page 4, lines 101-103). We are pulling together our longitudinal data spanning multiple time points from the same cohort in which we will be able to consider the standard definition.

2. Line 98: please explain $\geq 5\%$ with potential ambiguity. I assume any drug resistance mutation present at $\geq 5\%$ was included on the analysis. It would be interesting to know how many mutations were present $>20\%$ (Sanger consensus) and 5-20%

Response: Yes, the Reviewer is correct that any DRMs present at $\geq 5\%$ were included in the analysis. We revised this sentence in the Methods a bit for clarity (page 4, line 109). Additionally, our pipeline generated ambiguous consensus sequences using a 15% frequency threshold to align more closely with Sanger sequencing limitations, which were submitted to Stanford to assess drug resistance. As expected, fewer drug resistance mutations were detected in ambiguous sequences at the 15% versus 5% frequency threshold. Therefore, we presented 5% threshold data in our manuscript since using 15% threshold data can miss detection of important DRMs at VF.

Thirty-one of 70 samples (11 DTG and 20 non-DTG) were found to have drug resistance mutations called at a frequency of $\geq 5\%$ but not at a frequency $\geq 15\%$. IN Accessory mutations were observed in 3 DTG and 2 non-DTG samples, and one non-DTG sample exhibited an IN major mutation at the 5% threshold but not at the 15% level. No differences in PR mutations were observed among DTG patients, but differences in PR accessory mutations were identified in 3 non-DTG samples. Additionally, differences in major PR mutations were observed in 2 non-DTG samples when comparing the 5% and 15% thresholds. Differences in NNRTI mutations between these thresholds were found in 8 DTG and 7 non-DTG samples, while differences in NRTI mutations were found in 5 DTG and 9 non-DTG samples.

3. The 20% failure rate for genotyping is mentioned as a limitation; was this random or linked to pVL, collection period, collection site, ...?

Response: We thank the Reviewer for raising this concern. Our rate of successful genotyping was comparable to other recent reports, as mentioned in the Discussion. Unsuccessful genotyping in 20 participants was likely due to lower pVL levels and/or mismatches between the primer and target sequences. There was no evidence to suggest that failed attempts occurred due to differences in the way specimens were obtained (timing, site), initially processed, stored, shipped or handled downstream in the sequencing process. Likely reasons for failed genotyping have been added to the Discussion (pages 7, lines 257-258).

4. Line 207: it is unclear to which guidelines this is referring to (ref 27), are those the Liberia or US guidelines? **Response: This was referring to the US guidelines, which upon re-review, we decided was not very relevant to the situation in Liberia. Therefore, we have removed this citation.**

5. Line 210: was genotyping attempted for any of the samples with VL 40-999 copies/mL?

Response: Unfortunately, we only performed GRT on those who met our definition of VF, which was in line with the WHO's recommendation to perform resistance testing on those with >1000 copies/mL in LMIC (<https://iris.who.int/bitstream/handle/10665/372838/9789240076662-eng.pdf?sequence=1>). As a result, we do not have any data on participants with low-level viremia, which we cite as a limitation in the Discussion (page 7, lines 259-260).

6. Line 210: was low-level viraemia more common in the INSTI-group compared to the PI and NNRTI-group? Also it would be interesting to know if LLV is more common in PWH who initiated DTG compared to those who switched to DTG from a previous regimen.

Response: We thank the Reviewer for raising these points. Table 1 shows that 9.3% of those on EFV- or NVP-based regimens v. 12% of those on DTG-based regimens had LLV at baseline. The percentages of LLV for those who started ART on a DTG-based regimen vs. those who switched from NNRT- to DTG-based regimens were 13% v. 11%. These differences were not statistically significant. We have added a sentence in the Results to highlight these points (pages 4-5, lines 140-142). We also made some other minor changes related to LLV in the text to shorten our overall word count (page 4, line 139; page 7, lines 250-260).

7. There has been a lot of discussion on the importance of the denominator when reporting DTG resistance. Since you have a nice cohort with the number of people on DTG treatment known, it would be useful to add the prevalence of DTG resistance among all patients in DTG (suppressed and not suppressed) in addition to the currently reported DTG resistance prevalence among PWH and non-suppression.

Response: We thank the Reviewer for their input here. We discussed this point at length among our co-authors and the majority felt comfortable reporting this only in the context in which it was tested, which was among those with VF in whom virus resistance testing was successful. For this reason, we would favor reporting this as we have done.

8. Table 1: Were some patients in NFV?

Response: A total of 3 participants were on NFV-based ART. We have added a footnote to Table 1 to clarify this fact.

9. Table 4: I would suggest to only include ARVs that are currently in use.

Response: We thank the Reviewer for this suggestion. Our population includes participants who had been on ART for an average of 6.1 years. Thus, this is a fairly

treatment-experienced cohort, where many received older antiretroviral drugs such as zidovudine, didanosine, stavudine, and nevirapine that were commonly used as initial therapy in Liberia. We think inclusion in this table of resistance to these older drugs would be important since it has been demonstrated that accumulation of certain NRTI-associated mutations may reduce susceptibility to tenofovir, thus making VF more likely to occur. In addition, based on feedback from in-country stakeholders, we believe that having resistance data for some ARVs that might be introduced or recycled in the future as part of salvage ART regimens would be important to include. If agreeable to the Reviewer, we would prefer leaving the table as is.

Reviewer #3 (Remarks to the Author):

This is a very good work. It touches on a very important area. I have reviewed the manuscript and I am attaching a tracking version of the manuscript. The data analysis used is sound and valid and the reproducibility of the current work is very likely.

Line 28: The abstract contains quite a number of abbreviations that needs to be fully define. The authors should have a look at those that were not fully defined.

Response: We thank Reviewer #3 for carefully reviewing our manuscript and for their helpful suggestions. Our original abstract included 200 words to conform to the author instructions. Now we have slightly expanded some parts as suggested by this Reviewer's comments from the tracked version of the manuscript, including fully defining all abbreviations (page 2).

Line 29: Background is too vague. Needs to be expanded to give a better perspective of what the study is about.

Response: Done.

Line 32: This does not give a better perspective of what was done. This will make it difficult to appreciate the findings since there is no information on the subjects (numbers) that was used. The authors should elaborate more on this section

Response: Done.

Line 34: Information of the basic demographic data should be included in the findings to give readers who the participants were.

Response: Basic demographic data, including gender and age, has been added to the abstract to better characterize our participants (page 2, lines 38-39).

Line 38: Define HIVDR and please define all abbreviations when they are being used for the first time.

Response: More information about the method used to test for HIV drug resistance has been added to the Methods (page 2, lines 35-36).

Line 42: Conclusion?

Response: Interpretation was changed to Conclusion (page 2, line 49).

Line 44: Define InSTIs fully. Again, this has not been linked to dolutegravir anywhere in the abstract so why is it appearing here for the first time.

Response: InSTIs are now fully defined and linked to dolutegravir in the first sentence of the Background (page 2, lines 30-31).

Line 75: A summary of some of the methods stated here needs to be incorporated into the abstract section to give a better picture of what was done.

Response: Done (as mentioned above).

Line 91: What other clinical laboratory tests were performed and did they play any role in this study? If yes, they need to be listed otherwise delete it.

Response: As instructed, the reference to “other clinical laboratory tests” has been deleted (page 4, line 101).

Line 104: What class were those with pVL 40-999 classified as? Then again what role did they play in this whole work since the focus seems to have been on those with pVL <40 and those > 1000?

Response: We thank the Reviewer for this helpful comment. We have made some revisions to the Results to specifically identify this group with “low-level viremia (LLV)” and to highlight similar prevalence of LLV among participants receiving different ART regimens (pages 4-5, lines 139-142). Although the LLV group was not the focus of this study, their inclusion is important since they represent 12% of our total cohort and may be at higher risk of progression to VF and accumulation of drug resistance mutations compared to those with fully suppressed viremia, as we point out in the Discussion (page 7, lines 250-255).

Line 123: What was the reason for measuring D-dimer? It is not mentioned anywhere in the results section as to what the results was.

Response: In a previous study of sub-populations enrolled on two Ebola-related research protocols in Liberia, we found that HIV+ participants were over 6x more likely to die and had significantly higher d-dimer levels than HIV- participants (Moses SJ et al., PLoS One, 2021). Although we showed that the association between d-dimer and mortality was not restricted to individuals with HIV in that analysis, we were interested in including this potentially useful biomarker in the current study to determine whether there was an association with virologic failure. As the Reviewer points out, we failed to mention this lack of an association in the Results, which we have now corrected (page 5, line 149).

Line 177-182: The factors associated with virologic failure was not fully discussed. What were the reasons for these factors that were associated with virologic failure? Merely mentioning them is not enough.

Response: We are not able to conclude cause-and-effect relationships from our analyses, i.e., the reasons these factors were associated with VF. However, per the Reviewer’s suggestion, we expanded the discussion of these results in the context of other published reports (page 6, lines 207-212).

Line 196-203: These are just repetition of results and not discussion. Please revise.

Response: The aim here was to further discuss the participants with DTG resistance in more detail to emphasize the important likelihood that NRTI resistance predated DTG exposure, leading to DTG functional monotherapy. To more efficiently make this point, we moved the description of the two cases with DTG resistance to the Results (page 5, lines 184-187) and revised the remaining text in the Discussion (pages 6-7, lines 228-238).

Line 234-238: The factors associated with virologic failure is missing in the conclusion.

Response: We have now revised the last paragraph of the Discussion to include the factors associated with VF (page 7, lines 278-279).

Point-by-Point Responses to Referee Comments

Response: We thank the reviewer for their favorable response. We have included the text in the letter in the relevant sections of the manuscript, specifically, Line 246-8, 250-3 (Methods), and 311-19 (Results).

Response: We thank the reviewer for their favorable response.

Response: We thank the reviewer for their favorable response.